# Time and Mode of Epidemic HCV-2 Subtypes Spreading in Europe: Phylodynamics in Italy and Albania

**DOI:** 10.3390/diagnostics11020327

**Published:** 2021-02-17

**Authors:** Erika Ebranati, Alessandro Mancon, Martina Airoldi, Silvia Renica, Renata Shkjezi, Pranvera Dragusha, Carla Della Ventura, Anna Rita Ciccaglione, Massimo Ciccozzi, Silvia Bino, Elisabetta Tanzi, Valeria Micheli, Elisabetta Riva, Massimo Galli, Gianguglielmo Zehender

**Affiliations:** 1Department of Biomedical and Clinical Sciences “L. Sacco”, University of Milan, 20157 Milan, Italy; erika.ebranati@gmail.com (E.E.); martina.airoldi@uni-wuerzburg.de (M.A.); silvia.renica@unimi.it (S.R.); carla.dellaventura@gmail.com (C.D.V.); massimo.galli@unimi.it (M.G.); 2CRC-Coordinated Research Center “EpiSoMI”, University of Milan, 20122 Milan, Italy; 3Unit of Microbiology, Hospital Sacco of Milan, 20157 Milan, Italy; alessandro.mancon@live.com (A.M.); valeria.micheli@asst-fbf-sacco.it (V.M.); 4Faculty of Medicine and Surgery, Catholic University “Our Lady of the Good Counsel”, 1001 Tirana, Albania; renata_sh@libero.it (R.S.); p.dragusha@unizkm.al (P.D.); 5Viral Hepatitis Unit, Department of Infectious, Parasitic and Immune-Mediated Diseases, Istituto Superiore di Sanità, 00161 Rome, Italy; annarita.ciccaglione@iss.it; 6Unit of Medical Statistics and Molecular Epidemiology, University Campus Bio-Medico of Rome, 00128 Roma, Italy; m.ciccozzi@unicampus.it; 7National Institute of Health, 1001 Tirana, Albania; silviabino@gmail.com; 8Department of Biomedical Sciences for the Health, University of Milan, 20133 Milan, Italy; elisabetta.tanzi@unimi.it; 9Laboratory of Virology, Campus Bio-Medico University, 00128 Rome, Italy; e.riva@unicampus.it

**Keywords:** epidemiological history of HCV-2, HCV-2 subtypes, evolutionary demography of HCV-2, phylodynamics of HCV-2 in Italy and Albania, HCV-2 Re estimation

## Abstract

Hepatitis C virus (HCV) genotype 2 causes about 10% of global infections and has the most variable circulation profile in Europe. The history of “endemic” HCV-2 subtypes has been satisfactorily reconstructed, instead there is little information about the recent spread of the “epidemic” subtypes, including HCV-2c. To investigate the origin and dispersion pathways of HCV-2c, 245 newly characterized Italian and Albanian HCV-2 NS5B sequences were aligned with 247 publicly available sequences and included in phylogeographic and phylodynamic analyses using the Bayesian framework. Our findings show that HCV-2c was the most prevalent subtype in Italy and Albania. The phylogeographic analysis suggested an African origin of HCV-2c before it reached Italy about in the 1940s. Phylodynamic analysis revealed an exponential increase in the effective number of infections and Re in Italy between the 1940s and 1960s, and in Albania between the 1990s and the early 2000s. It seems very likely that HCV-2c reached Italy from Africa at the time of the second Italian colonization but did not reach Albania until the period of dramatic migration to Italy in the 1990s. This study contributes to reconstructing the history of the spread of epidemic HCV-2 subtypes to Europe.

## 1. Introduction

Hepatitis C virus (HCV) is one of the main causes of viral hepatitis, which first presents as an acute infection but evolves into chronic disease in 50–70% of cases [1]. The World Health Organization (WHO) has estimated that, in 2015, there were 71 million people living with chronic HCV infection at risk of developing cirrhosis and hepatocellular carcinoma (WHO data available at http://www.who.int/hepatitis/publications/global-hepatitis-report2017/en/, accessed on 16 February 2021).

HCV belongs to the positive-sense RNA virus family Flaviviridae, and is characterised by a high degree of genetic variability because it has a high replication rate and is under selective pressure from the immune system, and the error-prone viral protein NS5B, an RNA-dependent RNA-polymerase, lacks proof-reading activity [2]. This has led to the existence of at least eight main genotypes, more than 80 subtypes and some recombinant form, as stated by the International Committee for the Classification of Viruses (ICTV) in May 2019 (https://talk.ictvonline.org/ictv_wikis/flaviviridae/w/sg_flavi/634/table-1-confirmed-hcv-genotypes-subtypes-may-2019, accessed on 16 February 2021) [3].

The geographical distribution of the genotypes and subtypes reflects the diversity of the transmission rates and routes in different areas: epidemic strains such as 1a, 1b, 2a, 2b, 2c and 3a spread throughout the world as a result of the use of blood products, invasive procedures and syringe sharing by intravenous drug users, and account for the majority of infections in western countries, whereas endemic subtypes have low transmission rates and generally unknown modes of transmission, thus leading to the local restriction and diversification typical of sub-Saharan Africa and south-east Asia [4,5,6].

Genotype 2 has been responsible for about 10% of global infections and has the most variable circulation profile in Europe: HCV-2a and -2b are the most prevalent subtypes in all of the major European countries (such as Germany, Great Britain and Sweden); subtype 2c is prevalent in Italy and Estonia; in addition to subtypes 2a, 2b, 2c, new and mixed subtypes have been isolated in The Netherlands, France and Belgium [7].

This diversified distribution makes it fascinating to investigate the molecular evolution of HCV-2 [8,9] because, although the evolutionary and social history of the “endemic” genotype has been satisfactorily reconstructed [10], little is known about the recent spread of the “epidemic” subtypes, particularly HCV-2c. Furthermore, there are some European countries (such as Albania) for which there are hardly any subtype distribution data, although the few studies that have been carried out [11,12] indicate the significant circulation of HCV-2 in particular of subtype 2c.

Direct-acting antivirals (DAAs) have recently become the new standard-of-care for the treatment of chronic HCV infections, and their >90% efficacy has induced the WHO to launch a program aimed at eliminating HCV as a public health threat by 2030 (WHO data available at http://www.who.int/hepatitis/publications/global-hepatitis-report2017/en/, accessed on 16 February 2021). The treatment regimen and the duration administration depend on the viral genotype and subtype, and so the global implementation of DAAs strategies requires a knowledge of the main HCV genotypes and subtypes circulating in different geographical areas. Given the particular nature of the Italian epidemiological picture (in which -2c is the most prevalent HCV-2 subtype) and the scarcity of data concerning the circulation of HCV-2 in Albania, the aim of this study was to characterize 245 new HCV-2 isolates in order to reconstruct the evolutionary history of HCV-2 in these two countries phylodynamically and phylogeographically.

## 2. Results

### 2.1. Phylogenetic Analysis of the Global HCV-2 Dataset

The 245 newly characterized HCV NS5B sequences of 208 Italian and 37 Albanian isolates were aligned with the reference sequences as described in Materials and Methods, and analyzed using a Bayesian approach. There were a number of significant clades (pp > 0.7) corresponding to the known HCV-2 subtypes, but all of the Italian and Albanian strains were included in the three major clades corresponding to the “epidemic” subtypes 2a, 2b and 2c (Figure 1).

The most prevalent subtype was HCV-2c in both Italy (201/208 isolates, 96.6%) and Albania (36/37 isolates, 97.3%). The other Italian isolates were classified as 2b (6/208, 2.9%) and 2a (1/208; 0.5%), and the single Albanian isolate outside the 2c clade, was 2a (1/37, 2.7%).

Table 1 shows the main demographic and behavioral risk characteristics of the Italian and Albanian patients infected with HCV-2.

More than 66% of the study population as a whole (163/245) were aged >60 years, but the mean age of the Italian patients was significantly higher than that of the Albanian patients (72 ± 15.2 vs. 50.5 ± 14.3 years; *p* < 0.005). There were more females than males among the Italian patients (61.1% vs. 38.9%) but this difference was not significant. Given that in the greater proportion of cases the risk of exposure was not known, in the remaining cases the iatrogenic exposure was the most frequent in both Italian and Albanian patients (Table 1).

### 2.2. Likelihood Mapping of the Main Dataset

The main dataset of 419 NS5B sequences used for the phylodynamic and phylogeographical studies was analyzed for phylogenetic signals using likelihood mapping (Tree-Puzzle 5.3.rc16). The percentage of dots falling in the central area of the triangle was 8.2%, thus indicating a fully resolved phylogenetic signal (Appendix A).

### 2.3. Phylogeographical and Phylodynamic Analyses

#### 2.3.1. Estimated Substitution Rates and tMRCA

The mean evolutionary rate of the dataset of HCV-2 NS5B sequences was estimated to be 2.4 × 10^−3^ subs/site/year (95%HPD 1.7–3.1 × 10^−3^) using the GTR+I+G substitution model, under constant population size coalescent and strict clock models selected as described above (see Section 4).

Figure 2 shows the dated tree including the patient and reference sequences of genotype 2 using the substitution rate estimated for the whole dataset.

Only five nodes in the internal backbone of the tree had posterior probabilities of >0.7 (A–E in Figure 2): nodes B, D and E corresponded to the MRCAs of the main “epidemic” HCV subtypes (2a, 2c and 2b respectively), whereas nodes A and C were in a deeper position and respectively represented a common ancestor of all of the HCV-2 strains, and a deeper ancestor of the HCV-2c and HCV-2a clades. Table 2 shows the estimated tMRCAs of the significant internal nodes.

The tree-root tMRCA was estimated to be a mean 435 years ago (YA: 95%HPD 223–684 YA), corresponding to the year 1574 (CI 1332–1793). The tMRCA estimates of the three main clades corresponding to nodes B, D and E were between 1948 (2b and 2c) and 1954 (2a). The estimated tMRCA of the highly significant node C preceding the common ancestor of HCV-2c was an average of 83.9 YA, corresponding to the year 1925 (95%HPD 1907–1954).

#### 2.3.2. Phylogeography of HCV-2

Figure 3 shows the phylogeographical Baysian tree indicating the most probable locations of the internal nodes by different branches colors.

The most probable location of the tree root was Guinea Bissau (st pp = 0.49 vs. France st pp = 0.32) (Table 2). The most probable locations of the MRCAs of the epidemic subtypes were Italy (st pp = 0.83) for HCV-2c, and The Netherlands (st pp = 0.99) for HCV-2b; Indonesia and The Netherlands had similar posterior probabilities of being the location of the HCV-2a ancestor (Indonesia st pp = 0.38; The Netherlands st pp = 0.24). The second highly significant internal node (C) of HCV-2c had Ghana as the most probable location (st pp = 0.99). The Albanian isolates tended to group into two sub-clades supported by low posterior probabilities (pp > 0.50).

Appendix A shows the 33 identified significant non-zero rates (BF > 3), most of which included Italy (10), Africa (8), France (5), and The Netherlands (4).

In particular, Italy showed highly significant linkages with Albania (BF = 4561), Argentina (BF = 86,827), The Netherlands (BF = 86,827), Venezuela (BF = 86,827), Tunisia (BF = 14,463), France (BF = 3766), and Ghana (BF = 29.5).

The phylogeographical reconstruction of migration flows (Figure 4a–d) showed that, after its origin in western Africa (Guinea Bissau) in the XVI century, the HCV-2 genotype migrated to central Africa, (in particular Ghana for HCV-2a and -2c). From Africa, it moved to Europe and had reached France, The Netherlands and Italy by the 1930s–1950s.

#### 2.3.3. Evolutionary Demography and R0/Re Estimates in Italy and Albania

The Italian and Albanian HCV-2c isolates were separated for further independent phylodynamic analyses. The skyline plot of the Italian sequences showed a growth in the effective number of infections during the period from the 1940s to the mid-1960s, followed by a plateau (Figure 5a). The direct estimate of HCV Re using the birth-death model and five intervals of time showed that it started to grow in the 1940s, reached a peak of 3.2 (95%HPD 1.1–7.5) in the period between the 1980s and early 2000s, after which it levelled off before starting to decrease only recently (Figure 5b). The model estimate indicated that the origin of the HCV genotype 2 epidemic in Italy was 70.5 YA (42.7–100.0 YA). The estimate of the becoming-non-infectious rate was 0.199 (CI = 0.02–0.47), corresponding to a mean time of infectiousness of five years (CI 2–50 years).

The skyline plot analysis of the Albanian sequences (Figure 5c) showed an exponential growth in the effective number of infections between the early 1990s and the mid-2000s, after which it reached a plateau. The birth-death model estimate of the trend of Re was similar: growth between the 1990s and early 2000s, followed by a rapid decrease. The origin of the epidemic was estimated to be 40.6 YA (26.3–86.9), corresponding to 1975 (Figure 5d).

## 3. Discussion

The aim of this study was to reconstruct the epidemiological history of HCV-2 and its “epidemic” subtypes in Europe, particularly HCV-2c as it is the most frequent HCV-2 subtype in Italy and Albania. To do this, we characterized NS5B sequences from 208 Italian and 37 Albanian isolates obtained from patients with chronic HCV-2 infection over a period of 25 years. A subset of these sequences was subsequently aligned with reference sequences obtained in different parts of the world at different times and retrieved from public databases in order to allow phylodynamic and phylogeographical analyses.

The vast majority (97%) of the isolates obtained from Italian patients were HCV-2c; the only exceptions were six isolates forming a significant monophyletic group of subtype 2b dating back to 1989, with an outgroup of six Dutch sequences obtained between the end of the 1980s and the early 2000s. This is in line with the findings of other studies showing that subtype 2c is very frequently found in Italy, whereas it represents only a minority of the isolates collected in other European countries [7]. Interestingly, the 2c strain also accounted for 97% of the isolates taken from HCV-2 positive Albanian patients. There are only a few studies of the prevalence of HCV genotypes in Albania, but all of them have found that HCV-2c is the most frequent subtype after HCV-1b, infecting 9–18% of all of the Albanian patients with chronic HCV infection [11,13,14].

Another main finding is that the Italian patients infected by HCV-2 were older than their Albanian counterparts: 66% were more than 60 years old. Other studies have found that HCV-2 is highly prevalent among elderly Italians [15,16,17]; it was also highly prevalent among Italian children, but its prevalence has been decreasing in those born after 1990 [18]. A number of epidemiological studies suggested that HCV-2 used to be even more prevalent in Italy in the past [15,16,17,19,20,21,22,23], which would explain its widespread prevalence among elderly Italians, but it has now been overtaken by other genotypes in younger subjects, who have mainly been infected as a result of intravenous drug use or contaminated blood transfusions [24].

The penetration of HCV-2 in Albania is probably more recent (see below), which would explain its greater frequency among younger Albanians with HCV-2 infection.

These data are in line with the rapidly changing epidemiology of HCV genotypes in Europe [25].

Several authors have tried to reconstruct the origin and global spread of genotype 2 using various phylogenetic approaches. One of the most important studies is that of Markov et al. [26], who suggested that HCV-2 originated in western Africa (Guinea Bissau) between 1400 and 1500 AD, and then migrated eastward to central Africa. In a subsequent study, the same authors demonstrated the central role of European (particularly Dutch and French) colonialism in the spread of endemic HCV-2 subtypes throughout the New World, whereas the dissemination of the epidemic subtypes 2a, 2b and 2c did not occur until the XX century, and was due to the use of contaminated blood products or intravenous drug use [10], as shown by their clustering on the basis of the mode of transmission. Other authors have confirmed the western African origin of HCV-2 and underlined the role of colonialism in its spread [8,26,27].

These studies were mainly aimed at reconstructing the history of endemic subtypes, but they were less effective in assessing the history of epidemic subtypes, possibly because of differences in the time scales of the evolution of the two groups of subtypes. It has been shown that the time dependency of substitution rates leads to overestimates when timescales are short (as in the case of the epidemic strains of HCV) and underestimates when timescales are long (as in the case of the endemic strains) [28]; however, most of the studies cited above used the same NS5B fragment substitution rate (5 × 10^−4^ subs/site/year) for both endemic and epidemic subtypes. This substitution rate was independently estimated using HCV-1 isolates collected after an epidemic event due to accidental transmission caused by a single batch of anti-rhesus immunoglobulin in Ireland in 1978 [29], and it was used by Pybus et al. in their first reconstruction of the phylodynamics of HCV genotypes [30]. However, various factors can affect the substitution rate of viruses [31], including the epidemiological/ecological dynamics of the infection in the host population [32,33,34] and the main routes of transmission [35], two characteristics that clearly distinguish endemic and epidemic HCV geno/subtypes [4].

As the aim of this study was to reconstruct the molecular evolution of epidemic HCV-2 subtypes by including a large number of Italian and Albanian HCV-2c sequences newly characterized by us, we estimated the substitution rate on the basis of sampling years, and obtained a mean value of 2.4 × 10^−3^ subs/site/year (CI 1.7–3.1 × 10^−3^ subs/site/year). This is faster than the mean rate used by other authors, but in line with a recent review of HCV evolution that suggested a mean evolutionary rate of about 1 × 10^−3^ subs/site/year [36]. It also confirms the observation that different genotypes/subtypes may have different substitution rates as HCV-2 has evolved much more rapidly than the other genotypes [37].

Despite our use of a faster rate, our phylogeographical reconstruction confirmed that the most probable origin of HCV-2 was western Africa (Guinea Bissau) at least 500 years ago, after which it spread eastward through central Africa to Ghana, an important center of its further spread out of Africa [26,27]. Our estimates of the tMRCAs of the significant clades and analyses of the significant linkages between locations suggest that the epidemic subtypes migrated to Europe between the 1930s and 1950s, when HCV-2 reached The Netherlands, Italy and France, which became the main centers of the further worldwide spread of the epidemic up to the 1980s.

Our phylogeographical analysis indicates that HCV-2c has two internal node MRCAs: one dating back to the 1950s with a more probable location in Italy, and another (including also HCV-2a subtype) dating back to the 1930s with Africa (i.e., Ghana) as the most likely location. Interestingly, other authors have suggested Ghana as a further possible location of the origin of HCV-2 [27], thus underlining the central role of the country in its dissemination. The analysis of genetic flow rates suggests that HCV-2c moved to Italy between the 1930s (when its ancestor existed in Africa) and the 1950s (when the common ancestor of the currently circulating HCV-2c appeared in Italy), after a largely unknown journey lasting at least 20 years.

The main obstacle to reconstructing the route taken by this original strain before its arrival in Italy is the lack of isolates from other African countries. Nevertheless, as in the case of the other HCV-2 subtypes, it is reasonable to assume that European colonial history played an important role in the arrival of HCV-2c in Italy [38], even if the history of Italy in Africa is more restricted in time and space than that of other western countries: from the 1890s to the end of World War II in regions of eastern Africa (Somalia, Eritrea, Ethiopia) and northern Africa (Libya). One recent study has shown a high prevalence of HCV-2c in Ethiopia [39], where Italian soldiers and often resident migrant workers were present from 1936 to 1941. It is therefore possible that, after the end of World War II, the Italians who returned to Italy carried HCV-2c infection with them. Further studies including more African isolates would be useful to clarify how HCV-2c reached Ethiopia from Ghana, and the role played by eastern Africa in allowing HCV-2c to reach Italy.

The Bayesian skyline plot of population dynamics showed that, like that of other HCV subtypes in other western countries [10,40], the exponential growth of HCV-2c infection in Italy started in the 1950s and continued until the 1980s. We also traced the changes in the Re of epidemic HCV-2c in Italy using a birth-death model to estimate the effective replication number and other important epidemiological parameters. The results of these analyses indicate that, over a period of about 70 years (the estimated origin of the epidemics), Re rapidly increased between the 1950s and the 1980s, continued to grow at a slower rate until the 2000s, and then started to decline.

On the contrary, the skyline analysis showed that exponential growth of both the effective number of infections and the Re in Albania started later, between the early 1990s and the 2000s- Re began to decrease in 2000s. Albania was associated with Italian colonization during the twenty years of the fascist period, but our estimates of the tMRCAs of the significant Albanian clades suggests that HCV-2c reached Albania from Italy more recently, between the 1990s and 2010. During that period, Albania was going through a political and economic crisis due to the fall of the communist regime, and hundreds of thousands of Albanian migrated to Italy [41], particularly in the early 1990s and the late 1990s/early 2000s. This suggests that the infection was exported from Italy to Albania when infected Albanian migrants returned to their native country.

In conclusion, the findings of this study (the first to report the trend of the effective reproductive number of HCV-2 in two European countries) contribute to reconstructing the history of the spread of epidemic HCV-2 subtypes to Europe in the XX century, and show that Italy was an important center of the spread of HCV-2c in the Mediterranean area.

## 4. Material and Methods

### 4.1. Ethics Statement

This retrospective research was conducted on serum samples collected for clinical purposes and stored in the participating centers. All data used in the study were previously anonymized, according to the requirements set by Italian Data Protection Code (leg. Decree 196/2003) and by the General authorizations issued by the Data Protection Authority.

Approval by Ethics Committee was deemed unnecessary because, under Italian law, such an approval is required only in the hypothesis of prospective clinical trials on medical products for clinical use (art. 6 and art. 9, leg. Decree 211/2003). Written informed consent for medical procedures/interventions performed for routine treatment purposes was collected for each patient.

### 4.2. HCV-2 Positive Patients and Sequences

The 245 HCV-2 NS5B sequences newly characterized in our laboratory were obtained from 208 Italian samples taken from patients attending the Infectious Diseases Department of L. Sacco University Hospital in Milan or the Virology Unit of the Campus Bio-Medico University in Rome, and 37 samples taken from Albanian patients attending the National Blood Transfusion Centre of Tirana. The data concerning the samples and patients (age, country of birth and town of residence) were collected between 2001 and 2016. Patients plasma samples with >1000 IU of HCV RNA/mL were genotypically analyzed using a VERSANT^®^ HCV Genotype 2.0—LiPA kit (Siemens Healthcare, Erlangen, Germany), and the same samples were further characterized by means of NS5B gene sequencing.

### 4.3. Sample Processing and HCV RNA Sequencing

Viral RNA was extracted from the patients’ serum (200 μL stored at −80 °C) using NucleoMag 96 Virus (Macherey-Nagel, Düren, Germany) and automated KingFisher™ Magnetic Particle Processors (Thermo Fisher Scientific Inc., Waltham, MA, USA) in accordance with the manufacturer’s instructions. Serum samples from healthy subjects were used as negative controls. The RNA was eluted in 50 μL of nuclease-free distilled water, and reverse transcribed using the SuperScript III reverse transcriptase protocol (Thermo Fisher Scientific, Waltham, Massachusetts, USA): the cDNA was amplified by means of nested polymerase chain reaction (PCR) using GoTaq^®^ DNA Polymerase (Promega, Madison, WI, USA). The primers for the first and second rounds of NS5B amplification and the PCR conditions have been previously described [42,43].

The fragments obtained by means of PCR were purified using a commercial purification kit (QIAquick PCR Purification Kit, Qiagen, Hilden, Germany), and then sequenced bi-directionally using a BigDye Terminator Kit version 3.1 (Applied Biosystems, CA, USA) in accordance with the manufacturer’s instructions.

The sequencing products were purified from a 10 μL sample by means of precipitation in an ethanol/sodium acetate mixture. Finally, the sequences were determined using an automated DNA sequencer (ABI PRISM 3130 XL Genetic Analyser, Applied Biosystems).

### 4.4. HCV-2 NS5B Datasets

The 245 newly characterized NS5B sequences were aligned for phylogenetic analysis with 273 reference sequences (representative of previously described HCV-2 subtypes 2a, 2b, 2c, 2e, 2f, 2i, 2j, 2m and 2q) retrieved from public databases (http://www.ncbi.nlm.nih.gov/genbank/, accessed on 16 February 2021).

Bayesian phylogeographical and phylodynamic analyses were carried out using a dataset of 419 sequences, including a total of 112 Italian NS5B sequences randomly selected from the main dataset and all of the Albanian sequences, which were aligned with 270 reference sequences. The reduction in the number of Italian sequences was necessary in order to limit the effects of sampling errors due to the number of taxa per location on the assessment of the posterior probability of the root location. The reference viral sequences were selected on the basis of the following inclusion criteria: (1) they had been published in peer-reviewed journals; (2) there was no uncertainty about the sub-genotype assignment of each sequence and all were classified as non-recombinant; and (3) the city/state of origin were known and clearly stated in the original publication. The accession numbers, sampling localities and characteristics of the isolates included in the dataset are summarized in Appendix A. The sampling dates ranged from 1985 to 2016.

Two other datasets were constructed: a subset including 199 Italian HCV-2c NS5B partial sequences, and a subset including 36 HCV-2c NS5B partial sequences collected in Albania.

The sampling locations of the isolates included in the main dataset were Albania (AL, *n* = 37), Argentine (AR, *n* = 26), Burkina Faso (BF, *n* = 5), Canada (CA, *n* = 3), Cameroon (CM, *n* = 7), China (CN, *n* = 10), Estonia (EE, *n* = 3), France (FR, *n* = 18), Ghana (GH, *n* = 18), Guinea Bissau (GW, *n* = 16), Indonesia (ID, *n* = 15), Italy (IT, *n* = 112), Morocco (MA, *n* = 9), Nigeria (NE, *n* = 9), The Netherlands (NL, *n* = 76), Russia (RU, *n* = 11), Spain (SP, *n* = 2), Suriname (SR, *n* = 8), Tunisia (TN, *n* = 15), the United States (US, *n* = 2), Venezuela (VE, *n* = 15), and Vietnam (VN, *n* = 2).

### 4.5. Likelihood Mapping Analysis

In order to obtain an overall impression of the phylogenetic signals in the partial NS5B gene sequences, we made a likelihood-mapping analysis [44] of 10,000 random sets of four sequences (quartets) generated using *TreePuzzle* software [45]. If the frequency of dots within the central area (star-like trees) of the triangle is more than 30%, it is likely that the data includes a high level of noise.

### 4.6. Bayesian Phylogenetic and Phylogeographical Reconstructions

The main dataset and the subsets were aligned using the *ClustalW* software included in *BioEdit* [46], followed by manual editing (final alignment length = 211 nucleotide).

The *JModelTest* was used to select the simplest evolutionary model fitting the data, which was the GTR+I+G model of nucleotide substitutions for the main dataset and the Albanian subset, and the GTR+G model for the Italian subset.

The phylogenetic tree, model parameters, evolutionary rates and population growth were co-estimated using the Bayesian Markov chain Monte Carlo (MCMC) method implemented in *BEAST* v.1.8.4 [47]. Statistical support for specific clades was obtained by calculating the posterior probability of each monophyletic clade. Four simple parametric demographic models (constant population size, and exponential, expansion and logistic population growth) and a piecewise-constant Bayesian skyline plot (BSP) under both a strict and a relaxed (uncorrelated log-normal) clock were compared as coalescent priors [47].

The phylogeographical reconstruction was made using the continuous-time Markov Chain (MCC) process over discrete sampling locations implemented in *BEAST* [48], and the Bayesian Stochastic Search Variable Selection (BSSVS) model that allows diffusion rates to be zero with a positive prior probability. Comparison of the posterior and prior probabilities that the individual rates would be zero provided a formal Bayesian factor (BF) for testing the significance of the linkages between locations: rates with a BF of >3 were considered well supported and assumed to be the migration pathway.

The HCV-2 dataset and the Italian HCV-2c subset were investigated by running two independent MCMCs for 500 million generations, with sampling every 50,000 generations; the Albanian HCV-2c subset was investigated using 50 million generations, with sampling every 5000 generations. The data were combined using *LogCombiner* v. 1.80 in the *BEAST* package. Convergence was assessed on the basis of the effective sampling size (ESS) after a 10% burn-in using *Tracer* v. 1.5 software (http://tree.bio.ed.ac.uk/software/tracer/, accessed on 16 February 2021). Only ESS’s of ≥200 were accepted. Uncertainty in the estimates was indicated by 95% highest posterior density (95% HPD) intervals, and the best fitting models were selected using the BF and marginal likelihoods implemented in *BEAST* [49]; in accordance with Kass [50], only 2lnBF values of ≥6 were considered significant. The trees were summarized in a target tree using the *Tree Annotator* program included in the *BEAST* package and selecting the tree with the maximum product of posterior probabilities (maximum clade credibility) after a 10% burn-in. The estimates of the time of the most recent common ancestor (tMRCA) were expressed as the mean number of years and 95% HPD before the most recent sampling dates (which corresponded to 2016). The final trees were visualized using FigTree v. 1.4 (available at http://tree.bio.ed.ac.uk/software, accessed on 16 February 2021). In order to visualize diffusion rates over time, it is also possible to convert the location-annotated MCC tree to a GeoJSON data format suitable for viewing with georeferencing software and, using the new SPREAD3 analytical tool, the MCC tree was converted to a JavaScript object notation (JSON) file. The visualization was rendered using a Data Driven Document (D3) library [51].

### 4.7. Birth-Death Skyline Plot Estimate of Effective Reproduction Number (Re)

In order to make a direct estimate of the changes in effective reproductive number (Re) over time, we reconstructed the phylogenetic tree of the Italian and Albanian HCV-2c isolates using a birth-death skyline plot [52], which makes it possible to infer the transmission rate (lambda), the death/recovery rate (delta), and the sampling proportion (p).

We used a serial birth-death skyline plot (a forward-in-time model allowing the parameters to change in a piecewise fashion) and a lognormal prior probability distribution of the reproductive number (R), with a mean value (M) of 0.0, a variance (S) of 1.25, and a total of five dimensions. The becoming-non-infectious parameter was estimated using a lognormal prior with M = 0.5 and S = 1.5, so that the 95% probability of the infectious period fell between 0.1 and 19.4 years, with a median of five years. The prior of the sampling probability followed a beta distribution with an alpha value of 1 and a beta value of 100, which corresponds to a minority of sampled cases in relation to all cases (from 0.0005 to 0.03).

As there are no data concerning the origin of the outbreak of HCV-2 infection in Italy and it is known that the circulation of HCV-2 has been widespread over the last 100 years, we used a lognormal prior with M = 4 and S = 0.4. The highest probability was a period between approximately 28.3 and 105 years ago, with a median of about 54.6 years.

Similar values were used for the serial birth-death skyline plot of the Albanian sequences, with the exception that there were four dimensions of Re and the prior for the origin was between 15 and 100 years ago.

## Figures and Tables

**Figure 1 diagnostics-11-00327-f001:**
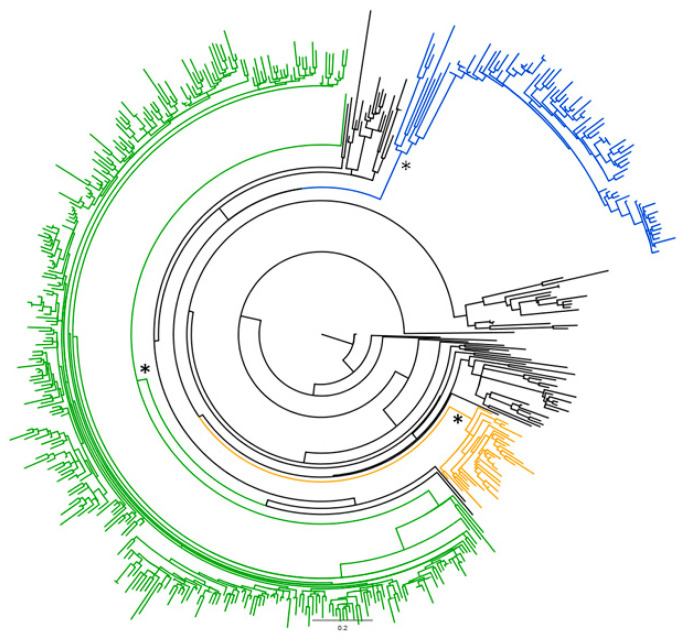
Bayesian phylogenetic tree of the 245 patient sequences and the 273 reference HCV-2 subtype sequences. The different HCV subtypes are highlighted in different colors (2a in yellow; 2b in blue; 2c in green: all of the endemic subtypes in black). *: Significant clades at ML and/or Bayesian analysis corresponding to epidemic subtypes.

**Figure 2 diagnostics-11-00327-f002:**
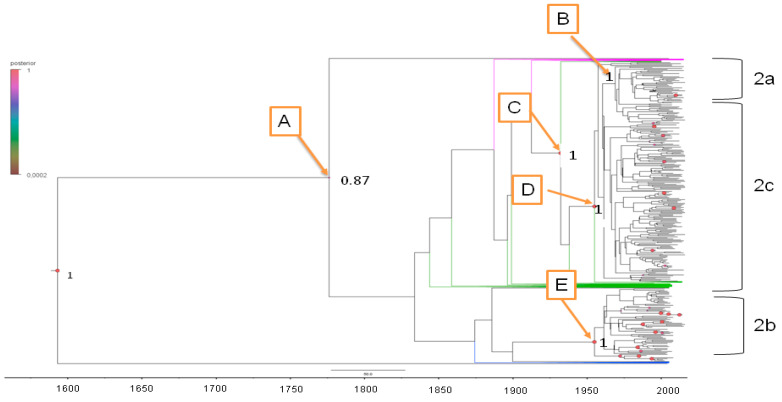
The maximum clade credibility (MCC) tree of the HCV-2 NS5B gene sequences. The letters and arrows indicate the significant internal nodes (posterior probability ≥ 0.8). The scale at the bottom of the tree represents calendar years. The clades corresponding to the three epidemic subtypes (2a, 2b and 2c) are highlighted.

**Figure 3 diagnostics-11-00327-f003:**
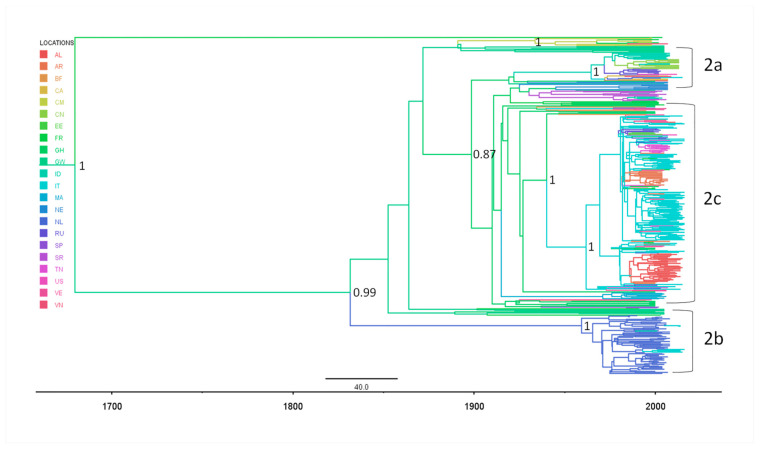
The Bayesian phylogeographical MCC tree of 419 HCV-2 NS5B gene sequences. The branches are colored on the basis of the most probable location of the descendent nodes (see color code in upper left inset). The initials on the internal nodes correspond to the country code of the most probable location, and the shapes of the internal nodes have been sized in proportion with their posterior probabilities. The scale at the bottom of the tree represents calendar years.

**Figure 4 diagnostics-11-00327-f004:**
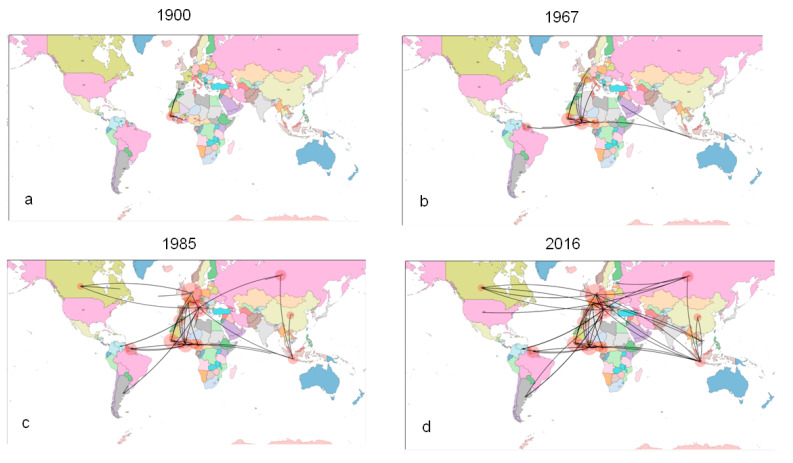
Significant non-zero migration rates of HCV-2 worldwide. Only the rates supported by a BF > 3 are shown. The map was reconstructed using SPREAD (see Section 4). The four panel (**a**–**d**) correspond to snapshots of the four different times indicated in correspondence with the panel.

**Figure 5 diagnostics-11-00327-f005:**
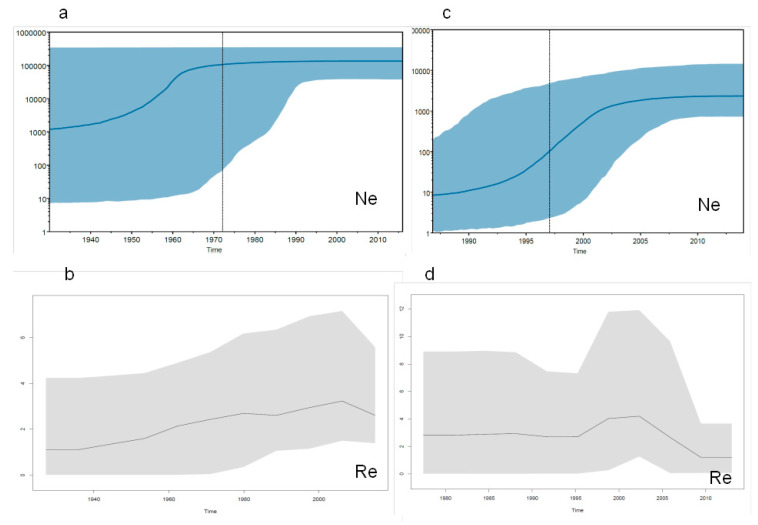
HCV-2 in Italy (**a**,**b**) and Albania (**c**,**d**). Bayesian skyline plots (**a**,**c**) and birth-death skyline plots (**b**,**d**) for the estimated median values and HPD intervals of effective population size (Ne) and effective reproduction numbers (Re).

**Table 1 diagnostics-11-00327-t001:** Demographic characteristics and reported risks of the study subjects with HCV-2 infection.

Characteristics	Italian Patients (208)	Albanian Patients (37)	Significant (p)
**Mean age (SD)-Years**	72 years (15.2)	50.5 (14.3)	<0.05
**Males proportion: absolute number (%)**	81 (38.9)	20 (54.1)	-
**Females proportion:absolute number (%)**	127 (61.1)	17 (45.9)	0.09
**Not-known risk-absolute number (%)**	158 (75.9)	31 (83.7)	0.3
**Iatrogenic risk-absolute number (%)**	34 (16.3)	6 (16.2)	-
**Other risks -absolute number (%)**	16 (7.6)	-	-

**Table 2 diagnostics-11-00327-t002:** Estimated times and credibility intervals (95%HPD) of the most recent common ancestors (tMRCAs) and most probable locations, with the state posterior probabilities (spp) of the main clades in the tree shown in Figure 3.

Node	Clade	tMRCA	Lower tMRCA	Upper tMRCA	Year Mean	Lower Year	Upper Year	Location	Location Probability
Root		434.7	223	684	1574	1332	1793	Gwinea	0.49
A		227.2	137	639	1781	1377	1879	Ghana	0.89
B	HCV-2b	61.03	49	75	1948	1941	1967	Netherlands	0.99
C		83.86	61.2	109.2	1925	1906.8	1954.8	Ghana	0.99
D	HCV-2c	60.9	44.2	79.7	1948	1936.3	1971.8	Italy	0.83
E	HCV-2a	56.2	39.4	73.1	1954	1942.9	1976.6	Indonesia	0.38

## Data Availability

The data presented in this study are available on request from the corresponding author. The data are not publicly available due to the patients’ privacy.

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
