# Peer review of "Time and Mode of Epidemic HCV-2 Subtypes Spreading in Europe: Phylodynamics in Italy and Albania"

_diagnostics, 2021, doi:10.3390/diagnostics11020327_

Round 1
Reviewer 1 Report
The authors present an interesting study on the chacterisation of hepatitis C virus HCV isolates from Italy and Albanian. Through the charaxterisation of NS5B sequences of these isolates, the authors have investigated various evolutionary aspects of the HCV. Overall, the study is well described and the conclusions drawn are well supported by the presented data.
My only concern with the paper is the current format of the abstract. I would suggest being rewritten along the lines of a more conventional structure:
- starting with a couple of sentences of background.
- What is the problem being addressed/investigated?
- Brief description of how it is addressed.
- Described what are the key results (most important ones only, not all)
- What are the conclusions of the study.
- Why are they important/what do they add to the field of research?
The current abstract basically summarises the study.
Minor suggestions
Page 2
Line 6 suggest "lines"
Page 8 Line 5 suggest deletion of "newly"
Author Response
We would like to thank the Referee for his/her helpful suggestions which contributed to improve our manuscript.
- We have now rewritten the abstract according to the indications of the referee.
- Minor changes suggested have been made to the manuscript.
Reviewer 2 Report
I read with interest the manuscript of Erika Ebranati et al entitled “Time and mode of epidemic HCV-2 subtypes dispersal in Europe: phylodynamics of HCV-2c in Italy and Albania”. the aim of this study was to characterise 245 new HCV-2 isolates in order to reconstruct the evolutionary history of HCV-2 in these two countries phylodynamically and phylogeographically.The study is methodologically sound. Introduction and discussion are exhaustive. Materials and methods are explained in detail, results are important. I have one question why data from neighbor to Albania countries were not included in the study.
Suggestions for manuscript correction are:
- Figures 1, 2 and especially figure 3 are of a bad quality. Its very difficult to see something.
- After full stops is missed space in many places: Paragraphs 2.3., 2.3.1., 2.3.2., 2.3.3., 4.1., 4.2., 4.3 and Page 2 after [11,12]
Author Response
We would like to thank the Referee for his/her helpful suggestions which contributed to improve our manuscript.
- The quality of figures 1, 2 and 3 have been improved. In particular, fig. 3 has been modified to increase the readability of the data.
- Changes suggested have been made to the manuscript.